# Chemical Composition of Nanoglobular Material on the Surface of Rubber Regenerate Prepared by Explosive Circulation Technology

**DOI:** 10.3390/molecules27217621

**Published:** 2022-11-07

**Authors:** Alexander Vasylievich Naumkin, Vyacheslav Mikhailovich Misin, Konstantin Igorevich Maslakov

**Affiliations:** 1A.N. Nesmeyanov Institute of Organoelement Compounds, Russian Academy of Sciences, 28 Vavilov Street, Moscow 119991, Russia; 2A.N. Frumkin Institute of Physical Chemistry and Electrochemistry, Russian Academy of Sciences, 31, Bld. 4 Leninsky Prospect, Moscow 119071, Russia; 3N.M. Emanuel Institute of Biochemical Physics RAS, 4 Kosygin Street, Moscow 119334, Russia; 4Department of Chemistry, Lomonosov Moscow State University, Leninskie Gory 1–2, Moscow 119991, Russia

**Keywords:** XPS, circulating explosive destruction, grinding, disintegration, rubber, nanoglobule

## Abstract

The rubber crumbs produced by the explosive circular destruction of worn-out automobile tires were studied. The crumbs showed high hydrophilicity. Their surface was analyzed by X-ray photoelectron spectroscopy. C, O, S, Zn, and Si were detected on the surface, and their chemical states were determined. The same chemical composition in the rubber crumb surface prepared by the explosive grinding of tires, as well as nanoglobules covering the crumb surface, was revealed. The appearance of polar groups on the crumb surface explains its high hydrophilicity and good compatibility with polymer matrices.

## 1. Introduction

Dealing with environmental problems involves not only the analysis of different types of pollutants [1] and their removal [2,3] but also the assessment and comparison of the advantages of various recycling technologies. The same is true for the disposal of used tires [4,5,6,7,8]. The total mass of worn-out and not recycled tires across the world is estimated to be ~17 million tons per year [9]. There are currently 3 billion tires stockpiled or landfilled in the EU and 1 billion in the US [10]. Due to the huge amount of waste tires, numerous studies are being carried out on the use of tire recycling products and, above all, crumb rubber in composite materials [11].

At the same time, the vast majority of worn-out tires are still not recycled but buried in landfills or stored on the surface of the lithosphere. The existing industrial methods of recycling of worn-out automobile tires have several disadvantages: (1) mechanical destruction (including freezing with liquid nitrogen) is energy-consuming; (2) incineration requires large investments into the purification of gaseous products; (3) liquid fuels produced by pyrolysis are of poor quality.

The unique technologies for the utilization of car tires by explosive milling were developed and adopted by the industry [12,13]. In this approach, nanoglobules are produced on the crumb surface, along with the formation of chemical functional groups.

These nanoglobules, 5–20 nm in size, are combined into clusters of various shapes and sizes, varying in the range of 100–1000 nm [14]. Nanoglobules are formed in two steps. In the first step, low-molar mass rubber decay products with unsaturated bonds are formed by the high-temperature destruction of tires [15]. Subsequently, these organic compounds condense on the rubber crumb surface to form oligomeric products, such as nanoglobules. Low-molar mass saturated and unsaturated linear, cyclic, and aromatic hydrocarbons were detected on the surface and in the sub-surface layers of all rubber crumb fractions [14]. In addition, 3–5 unit oligomers were found.

The rubber crumbs produced by explosive milling showed a good adhesion to polymer matrices, superior to that of rubber crumbs prepared by mechanical grinding. The former rubber crumbs are used as fillers in various matrices, which allow production of high strength composites [16]. The promising properties of the rubber crumbs produced by explosive milling can be explained by the presence of polar groups on its surface. Indeed, a positive effect of improving the performance of various hydrophobic carbon [17,18,19] and polymer [20,21,22] materials was observed when various polar hydrophilic groups were formed on their surfaces. For example, acrylic acid was grafted onto a polyethylene/polypropylene nonwoven fabric using γ-irradiation. The appearance of anionic COOH groups on the surface of the material was proven by physicochemical methods, including XPS [23]. Weft-knitted fabric produced from 100% meta-aramid fiber is widely used in different areas, such as the military, aerospace, electronics, and so on. It was successfully functionalized with cationic groups using the cationic polyelectrolyte poly (N,N-diallyl-N,N-dimethylammonium chloride [24]. The existence of oxygen containing groups on the surface of polyethylene modified with oxygen plasma was convincingly proven by various methods, including XPS [25]. Using XPS, the appearance of nitrile and anhydride groups on the surface of the carbon fiber after its chemical modification with tetracyanoethylene or maleic anhydride was proven [26]. The purpose of that work was to increase the strength of the fiber/epoxy resin composite by modifying the surface of the carbon fiber.

To increase the strength of composites using crumb rubber, its surface was modified by forming polar groups in various ways. An XPS study of tire rubber particles after plasma treatment showed the appearance of hydrophilic oxygen-containing functional groups, which improved the compatibility of the crumb with the asphalt matrix [27]. Oxidation of the crumb with a KMnO_4_ solution with sulfonation with a NaHSO_3_ solution resulted in the appearance of a large amount of hydrophilic hydroxyl and sulfonate groups [28]. Adding crumb rubber exposed to UV radiation to concrete reduced the strength of the concrete by only 6% [29]. An ATR-FTIR and XPS study of crumb rubber modified with ethanol plasma polymerization showed the presence of COOH, C–OH, and CHO hydrophilic groups on its surface, and its introduction into oil-well cement made it possible to increase its tensile and flexural strength [30].

Therefore, it is important to investigate the wettability of the crumb surface coated with nanoglobules and its chemical composition.

Different experimental methods can be applied to analyze the chemical structure and composition of tire recyclates; however, for many practical applications, the surface analysis is necessary because the chemical composition of the surface determines the physicochemical properties of materials, such as adhesion and compatibility with possible species of the future composite. As a surface-sensitive technique, X-ray photoelectron spectroscopy (XPS) allows us to determine the elemental and chemical compositions of the surface, and it is widely applied for the study of materials produced from used tires [4,7,31,32,33,34,35,36,37,38,39,40,41,42,43,44,45,46,47]. It is well known that tire recyclates contain a mixture of various substances whose compositions should be analyzed before their further use. The main elements present in tires are C, H, and O. The ground tires also contain S, Zn, Cl, and Si [4,32,38,40,46].

The proposed surface quality control by XPS can be used for the analysis of other types of regenerated rubber and may be of particular interest for the technology being developed for the destruction of large tires of quarry trucks with the obligatory use of regenerated products in industry. This approach may serve as an objective test that makes it possible to assess the environmental friendliness of the technologies being developed.

## 2. Materials and Methods

Rubber crumb samples RC 0-1, RC 1-3, RC 3-5, RC 5-10 with the average crumb sizes of 0–1, 1–3, 3–5, 5–10 mm, respectively, produced by “ZAO Tire recycling plant No. 1” (Raduzhny, Vladimir region) were studied in this work. Figure 1 shows photos of crumb samples of three fractions RC 0-1, RC 1-3, and RC 3-5. The surface topography of the RC 0-1, RC 1-3, RC 3-5, and RC 5-10 fractions were analyzed by atomic force microscopy [14]. It was found that they exhibited an amorphous structure without supramolecular ordering at the distance scale of ~1–40 nm with globules of 5–20 nm in size.

A sample of crumb fraction RC 3-5 was extracted with chloroform to remove highly soluble organic compounds present on the crumb surface. This sample is designated as RC 3-5 *. The technology of the tire destruction process is generally as follows. A stack of 10 tires is placed in a closed reactor and cooled to −70 °C. An explosive is placed inside this stack, which is distributed in height along the entire stack, as well as special technological additives. After the explosion, a technical mixture of crumb rubber of various sizes is formed, from which the metal cord is removed magnetically. The resulting rubber crumb is divided into fractions using a set of sieves with holes of different diameters. The main technological methods are given in patents [12,13].

Initially the samples were tested for wettability. The rubber chips were placed into water filled containers and mixed for 30 min for RC 0-1 and for 5 min for all other samples. During the stirring process, a part of the rubber crumbs settled to the bottom of the container, while the other part floated to the water surface. Each part was extracted, dried and weighed. The measured mass of the two parts coincided well with the mass of crumbs poured into the containers.

The XPS spectra were acquired on a Quantera SXM spectrometer (Physical Electronics, Chanhassen, MN, USA) with a monochromatic Al *K*_α_ radiation source (hν = 1486.7 eV, 25 W). The pass energies of the analyzer were 280 eV for survey spectra, 55 eV for high resolution XPS scans, and 140 eV for C KLL Auger spectra. The binding energy scale of the spectrometer was preliminarily calibrated using the position of the peaks for the Au 4f_7/2_ (83.96 eV), Ag 3d_5/2_ (368.21 eV), and Cu 2p_3/2_ (932.62 eV) core levels of pure metallic gold, silver, and copper. The diameter of the analyzed area was approximately 100 μm. Granules were mounted on a holder using a double-sided nonconductive adhesive tape and analyzed at room temperature at a pressure lower than 1∙10^−8^ Torr. The dual beam neutralizer was used, and the spectra were charge-corrected to give the C 1s component attributed to C−C/C−H bonds a binding energy of 285.0 eV. The inelastic background was subtracted by the Shirley method. Quantification was carried out in the MultiPak data reduction software (V.8.2).

## 3. Results and Discussion

### 3.1. Wettability of Rubber Crumbs

In experiment 1, stirring was performed for 30 min. In the following experiment 2, the crumb was allowed to stand in water for 12 h. The floating part of the crumbs was stirred for 30 min, and after settling for 20 min, it was removed, dried, and weighed. The test results for all rubber crumb samples are presented in Table 1.

In experiments 3–6, a portion of the crumbs quickly (within 1–3 min) settled to the bottom of the vessel after contact with water. No significant increase in the amount of settled crumbs was observed after further stirring for 5 min. In Experiment 1, crumb aggregates were observed on the water surface when the RC 0-1 crumbs made contact with water. It took 30 min to break up these aggregates and precipitate some crumbs in the bottom of the container. Interestingly, drying of these crumbs also took a lot of time. Optical microscopy images showed a large number of fine textile fibers in these rubber crumbs that originated from the tire cord. These fibers were close in size and suspendability to the finest rubber crumbs. As a result, the textile fibers were separated together with the RC 0-1 rubber crumbs. In the larger fractions of rubber crumbs, fine textile fibers were practically absent. The mixture of fine textile fibers and the RC 0-1 rubber crumbs has a lower density than water. As a result, the mass percentage of the floating part was increased up to 39.6% in experiment 1 (Table 1). In this case, the floating part of the mixture was evenly moistened with water within the thickness of the floating layer.

The results of the comparative test, in which rubber crumbs were prepared by mechanical grinding of tires using the roll technology according to Technical Conditions 38.108035-97, RD-0,8, (Chekhov Regeneration Plant, Chekhov, Moscow region, Russia), look quite different. The crumbs were poured into water and stirred for 60 min. No crumbs settled to the bottom were visually observed. After additional soaking in water for 12 h and stirring for 30 min, only 11% of the crumbs settled to the bottom.

This result can be explained by the nonpolar nature of the tire rubber surface and, therefore, its low wettability with polar solvents, such as water. The technology used for the production of the RD-0,8 crumbs does not lead to the chemical modification of the crumb surface and does not increase its wettability. High hydrophilicity of the RC rubber crumbs produced by the explosive grinding of tires ensures their efficient application in the composites with matrices containing polar groups [12,13]. Because of the high strength of the crumb–matrix bonds, the composite material does not contain cracks and micro-voids that can be concentrators of mechanical stresses. The same positive effect is expected for cement composites, which was confirmed by the results of studying the properties of real compositions. High hydrophilicity of rubber crumbs can be explained only by the presence of polar groups on their surface, which can be confirmed by XPS.

### 3.2. XPS Analysis

Rubber crumbs RC 0-1, RC 1-3, and RC 3-5 were analyzed by XPS in three areas **a**, **b** and **c,** selected using secondary electron images. These areas were designated as RC 0-1a, RC 0-1b, RC 0-1c, RC 1-3a, RC 1-3b, RC 1-3c, RC 3-5a, RC 3-5b, and RC 3-5c. Table 2 summarizes the XPS quantification data. The O (Si), O (Mg), and S (Zn) columns show the oxygen concentrations after subtracting the contributions from Si–O, Mg–O, and Zn–S bonds, respectively.

The high concentration of oxygen in the samples may be due to chemical reactions between gaseous oxygen and radicals formed as a result of mechanochemical destruction. Oxygenation of the obtained unsaturated fragments with C=C and C≡C bonds is possible [14,15]. The detection of the appreciable amounts of Si−O, O−Si(O)−O or SiO_2_, bonds is apparently explained by the existence of silicon oxide present on the surface of the destroyed rubber tires. A sufficiently high concentration of C−Si bonds was also detected, and it may be related to the interaction of carbon radicals formed during mechano-destruction with Si containing species. The presence of sulfur is associated with its use as a rubber vulcanizing agent, while a small amount of Zn is due to the use of various zinc compounds (zinc isopropyl xanthate, zinc dibutyl dithiophosphate, zinc dialkyl dithiocarbamate, etc.) as rubber vulcanization accelerators. Zinc may also be a part of the metal cord. In addition, Zn and Si could appear as a result of the use of zinc oxide and silicic acid as active fillers improving the mechanical characteristics of rubber [47,48,49]. Mg was detected in very small concentrations in two fractions of rubber crumbs in several analyzed areas. Mg oxides and polysulphides are also used in rubber mixtures as vulcanization accelerators. At the same time, accelerators show their highest activity in the presence of activators—metal oxides, in particular Zn, which were also detected [47,48,49].

Figure 2 shows the C 1s spectra of the rubber crumb samples. The spectra are normalized to the maximum intensity. To determine the presence of different carbon groups and their relative concentrations, the spectra were fitted with several components using the reference data on chemical shifts [50]. The minimum Gaussian width of the components was chosen from the fitting of the C 1s spectrum of sample RC 0-1c that showed the smallest full width at half maximum of the main peak. The fitting parameters for the C 1s spectra are presented in Table 3. The main differences in the spectra are observed in the relative concentrations of Si−C, C−C, C−O, and C=O bonds.

The peaks observed in C 1s spectra at 284.6–284.8 eV are sometimes attributed to the graphitized carbon [4]. However, the spectra of these carbon species typically show the binding energy of the main peak of 284.44 eV, the presence of the π–π * satellite at ~291.5 eV, [51,52], and the characteristic Auger lineshape that reflects the interaction between graphene layers [53]. Graphitized carbon can be detected both by Raman spectroscopy and XPS [33,34,42]. Hood et al. observed the sp^2^ carbon species by XRD and XPS [43]. The C 1s spectra of the rubber crumb samples (Figure 2) do not show satellites typical for aromatic carbons, which indicate the absence of these carbon species in the samples, at least in the amounts comparable to the detection limit of the technique. The contribution from O=C−O groups is absent in the C 1s spectra of samples RC 1-3b and RC 3–5 (areas **b** and **c**).

Carbon in the samples mainly forms C−C/C−H bonds (Table 3). Because the shape of the Auger spectra of hydrocarbon polymers depends on their chemical structure, we compared the C KVV Auger spectra of all the rubber crumb samples with the reference spectrum of polyethylene and graphite, which are identifiers of sp^3^ and sp^2^ states, respectively.

The C KVV Auger spectra of sample RC 0-1 (Figure 3) for areas **a** and **c** do not differ much from that of polyethylene, which points to a weak effect of oxygen on the outer layers of crumbs because the information depth of C KVV Auger electrons (36 Å) is approximately three times lower than that of C 1s photoelectrons (100 Å) [54]. The spectrum in area **b** is slightly different. A good coincidence of the C KVV Auger spectra with that of polyethylene is also observed for sample RC 1-3. The Auger spectrum of sample RC 3-5b is significantly different and close to that of polypropylene [50]. The reason for this fact is unclear and requires additional study. The Auger spectrum of sample RC 0-1b seems to be intermediate between spectrum of RC 3-5b and the spectra of other samples. A comparative analysis of all the photoelectron and Auger spectra allows us to summarize the data on the chemical states of carbon atoms and heteroatoms in the rubber crumb samples. Thus, the C KVV Auger spectra of the rubber crumb samples indicate the absence of sp^2^-species.

According to the XPS results (Table 3), approximately 14% of carbon atoms in the sample RC 3-5c are bonded to oxygen, which strongly contradicts the quantification data (Table 2), demonstrating an oxygen content of only approximately 7%. This fact leads us to reconsider the data in Table 2, taking into account that the quantification was made based on the atomic sensitivity factors (ASF). These ASF ignore the matrix effects, primarily associated with the inelastic mean free path of electrons (λ), which leads to much larger errors in the quantitative analysis than those resulting from the uncertainties in the component areas in the C 1s fitting procedure. In the case of C 1s component analysis, the quantification is based on practically the same λ values. Nevertheless, a comparative analysis of the data obtained by both methods makes it possible to reveal more reliable results.

Another possible reason of the contradiction is the possibility of C−S bond formation, both in the form of C−S and C−SO_2_. These bonds were not taken into account in the fitting model demonstrated in Figure 2 because of the low content of sulfur and the small chemical shifts of such bonds. The C−S bond induces a chemical shift in the range of 0.21–0.52 eV, while the C−SO_2_ bond leads to a shift of 0.31–0.64 eV [50]. On the other hand, the chemical shift used for the C−O bonds (about 1 eV) does not fall within the 1.13–1.75 eV range typical for such bonds [50], so it may be assigned to low-molecular fragments [55,56].

Nevertheless, we fitted the C 1s spectrum of sample RC 3-5c using an extra component attributed to the C−S state. The addition of this component led to a slight decrease in the relative content of oxygen determined from the C 1s spectrum. Thus, it can be concluded that a quantitative analysis using ASF underestimates the oxygen content. A similar phenomenon was observed under comparison of the quantification data for activated carbon [57]. Because the maximum sulfur concentration was observed in RC 0-1a, the same fitting model was used for this sample (Figure 4). In this case, the calculated O/C ratio decreased by 1.4 times but, nevertheless, remained 1.3 times higher than that calculated based on ASF. Because of the lower sulfur concentration, this effect was not so pronounced in sample RC 3-5c. Therefore, the extra component assigned to the C−S bond was not used for other sulfur-containing samples.

Figure 5 shows the S 2p spectra of the sulfur-containing samples. The spectra are fitted with two or three spin-orbit doublets with S 2p_1/2_–S 2p_3/2_ splitting of 1.2 eV and branching ratio of ½. The peak at 169.1 eV is characteristic of the sulfate group. Sulfates were unambiguously detected spectra only in samples RC 0-1a, RC 1-3b, RC 1-3c, and RC 3-5c, and their relative concentrations are higher in the sample RC 1-3c. The peaks at ~162 and 163.7 eV are assigned to C−S and Zn−S bonds, respectively.

The comparison of the Gaussian widths of the high-energy and low-energy components points to the contribution of at least two components to the low-energy peak. These components can be attributed to the Zn−S and C−S bonds. Figure 5 shows the result of fitting of the S 2p spectrum of RC 3-5c with three components centered at 162.7, 164.5 and 169.5 eV. Their intensity ratio is 17:34:49 and they can be, respectively, attributed to the Zn−S, C−S, and C−SO_x_ bonds. The same peak parameters (positions and widths) were used to fit other S 2p spectra. The relative contents of sulfur species in rubber crumb samples are presented in Table 4. It should be noted that the absence of zinc in sample RC 1-3a correlates with the absence of sulfur.

The binding energies of the Zn 2p_3/2_ peaks are practically the same for all samples and close to 1022.2 eV. It is known that the binding energy of the Zn 2p_3/2_ peak shows a poor correlation with the chemical state of Zn atoms. For ZnO, ZnS, and Zn (OH)_2_ these binding energies are in the ranges 1021.20–1022.5 eV, 1021.70–1022.0 eV, and 1021.8–1022.7 eV, respectively [51]. Therefore, the Auger parameter is commonly used to determine the chemical state of Zn atoms. However, the low concentration of zinc in the samples in our case, insufficient energy resolution, and overlapping of the Auger parameters of different Zn compounds (2009.1–2009.2 eV for Zn (OH)_2_, 2009.50–2011.0 eV for ZnO, and 2010.3–2011.9 eV for ZnS) [51] also makes it difficult to distinguish between the Zn−O and Zn−S bonds. However, based on the binding energy of the S 2p and O 1s components, one can identify Zn−O and Zn−S bonds. The O 1s spectra of the samples are broad enough to fit all the above-mentioned states into them. The signal in the O 1s spectra of all the samples above 534.0 eV is a fingerprint of water. Because of the variety of possible oxygen states in the samples, the unambiguous fitting of the O 1s spectra is impossible, and they were not fitted in our work.

Binding energies of the Si 2p components and the low-energy component in the C 1s spectra evidence the presence of siloxane groups. The asymmetry of some Si 2p peaks at the high-binding energy side indicates the presence of O−Si(O)−O or SiO_2_ groups. The analysis of the Mg 2p spectra revealed the oxidized state of Mg, which confirms its origin from the vulcanization accelerator.

Carbon atoms are mainly present in the form of alkanes. The C−O, C=O and O=C−O groups were identified in the C 1s spectra. The correlated analysis of the C 1s, S 2p, and Si 2p spectra allowed us to identify C−S and C−Si bonds. The latter are observed in a sufficiently high concentration. Therefore, C−O, C=O, O=C−O bonds strongly contribute to the total oxygen concentration. These results are not surprising as the interaction of oxygen with radicals and C=C bonds formed under explosive grinding of rubber by the mechanical deletion mechanism [12,13,15] is expected. The S 2p spectra display the presence of SO_x_ groups as well.

The use of sulfur as a rubber vulcanizing agent explains the presence of C–S bonds at concentrations comparable to oxygen ones. The oxidation of sulfur produces SO_x_ sulfo-groups on the surface. The detection of silicon, zinc, and magnesium atoms can be explained by various reasons, for example, by additives in rubber mixtures before their vulcanization [47,48,49].

Figure 6 shows the O 1s spectra fitted in accordance with the results presented in Table 2 and Table 3. The spectra fitted with Gaussian peaks assigned to physisorbed and chemisorbed H_2_O (H_2_O_ph_, E_b_~531 eV; and H_2_O_ch_, E_b_~534 eV), oxygen bonded to Zn, Mg, S, and Si (E_b_~530.8, 531, 531.7, and 532.0 eV) and to carbon in C=O, C-O-C, and C(O)O groups (E_b_~532.3, 532.8, 531.9, and 533.8 eV, respectively). It should be noted that a good correspondence with the quantification data and fitting of the C 1s spectra is observed (Table 5). The next very important point is the presence of physisorbed and chemisorbed water in the samples, despite the presence of the samples in an ultrahigh vacuum (H_2_O_t_ + H_2_O_ph +_ H_2_O_ch_). It is evidence of a strong interaction of water with the surfaces of the samples studied. A possible reason for this phenomenon may be the presence of hydrophilic sulfo- and oxygen-containing groups and solo polar bonds C–S, Mg–O, Zn–O, and Zn–S on the surface of the crumb rubber. Good wettability of all fractions of crumb rubber (Table 1), obtained by explosive circulation technology, is ensured by the combined action of both hydrophilic groups and polar bonds, which can be in different proportions in different fractions.

It was interesting to compare the surfaces of the original crumb and crumb treated with an extractant and not having an additional layer of nanoglobules on the surface. The analysis of the C 1s, O 1s, Si 2p, Zn 2p photoelectron, and the C KVV Auger spectra of sample RC 3-5 before (RC 3-5a, RC 3-5c) and after extraction with trichloromethane (RC 3-5 * showed that the spectra practically coincide. This fact indicates the same chemical composition of the surface of rubber crumbs and nanoglobules that cover this surface. This means that under explosive grinding of tires, almost identical chemical processes proceed both in nanoglobules and in the surface layers of rubber chips. Thus, the high hydrophilicity of rubber crumbs and the possibility of producing high strength composites are well explained by the XPS data. Indeed, the chemical modification of rubber crumbs under explosive grinding produces a large number of polar, hydrophilic oxygen-containing groups on the surface. These groups contribute to the high hydrophilicity of the rubber crumbs. Since the tire rubber density is higher than that of water (1.1–1.4 versus 1.0 t/m^3^) [56], up to 94% of the wetted rubber crumbs settle to the bottom. The presence of fine textile fibers in RC 0-1 reduces the density of the rubber chips. Therefore, a significant portion of the rubber crumbs did not settle to the bottom, though they are well wetted with water (Table 1).

## 4. Conclusions

A high hydrophilicity of the rubber crumbs produced by the explosive circular destruction of automobile tires was demonstrated and explained. The same chemical composition in the rubber crumb surface obtained by explosive grinding of tires and nanoglobules covering these rubber crumbs was observed. Carbon in the rubber crumbs was found to be mainly in the form of alkanes. Oxygen was mainly detected as C−O, C=O, and O=C−O bonds. The C−S bonds in concentrations comparable to the oxygen ones originated from the rubber vulcanizing agent. Magnesium, zinc, and silicon were also detected, and the reasons for their appearance were proposed. The approach used in this work to analyze the quality of the surface can be applied to various types of waste rubber regeneration products prepared by other technologies. The functional groups formed on the surface of the crumbs under explosive circular destruction of rubber tires contribute to their better compatibility with other components of composites without additional processing.

## Figures and Tables

**Figure 1 molecules-27-07621-f001:**
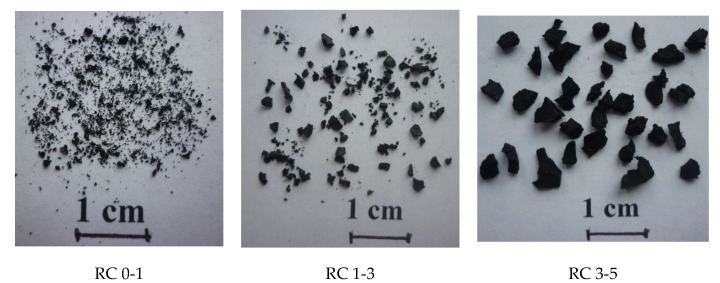
Photographs of crumb rubber fractions RC 0-1, RC 1-3, and RC 3-5.

**Figure 2 molecules-27-07621-f002:**
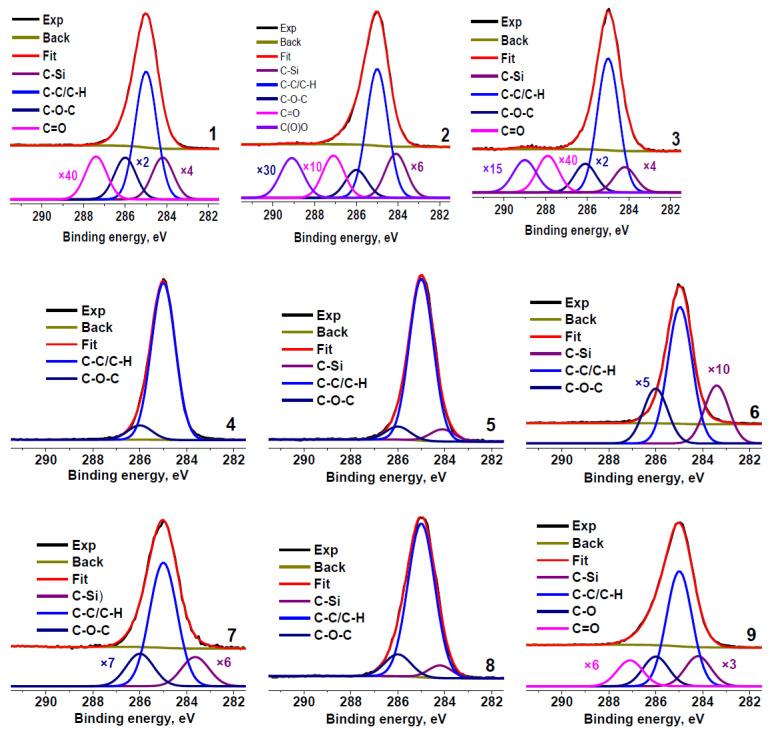
C 1s photoelectron spectra of rubber crumbs, measured in different areas: RC 0-1—**a** (**1**), **b** (**2**), **c** (**3**); RC 1-3—**a** (**4**), **b** (**5**), **c** (**6**); RC 3-5—**a** (**7**), **b** (**8**), **c** (**9**).

**Figure 3 molecules-27-07621-f003:**
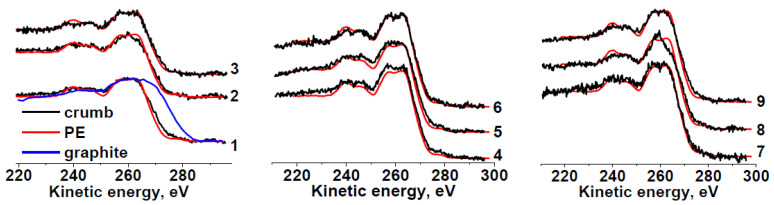
Comparison of C KVV Auger spectra of RC 0-1—**a** (1), **b** (2), **c** (3); RC 1-3—**a** (4), **b** (5), **c** (6); RC 3-5—**a** (7), **b** (8), **c** (9) with spectra of polyethylene (PE) and graphite.

**Figure 4 molecules-27-07621-f004:**
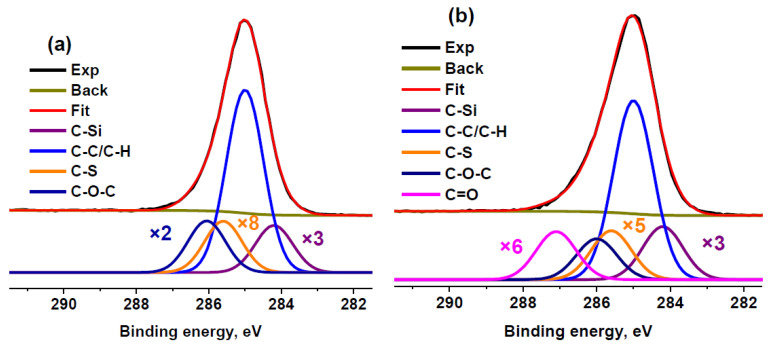
XPS C 1s spectra of RC 0-1a (**a**) and RC 3-5c (**b**) fitted with the extra component attributed to C–S bonds.

**Figure 5 molecules-27-07621-f005:**
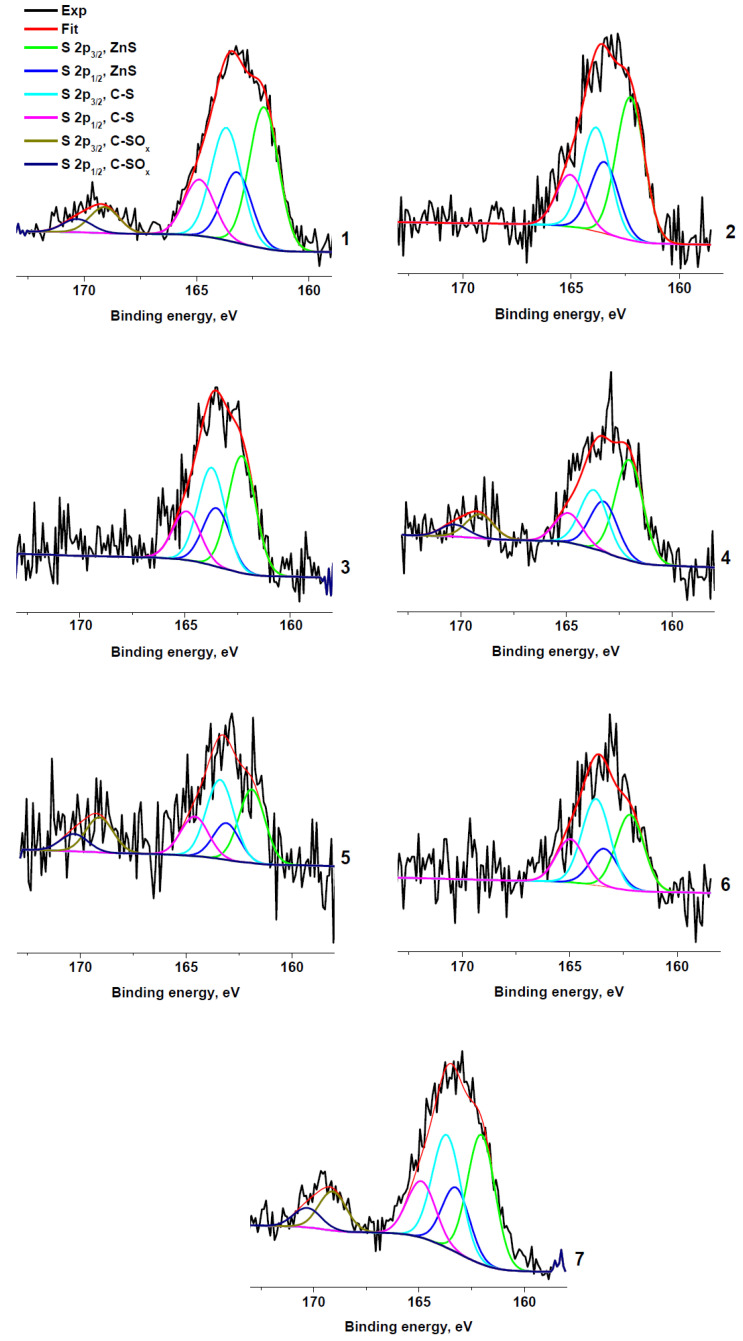
XPS S 2p spectra: RC 0-1a (**1**), RC 0-1b (**2**), RC 0-1c (**3**), RC 1-3b (**4**), RC 1-3c (**5**), RC 3-5b (**6**), and RC 3-5c (**7**).

**Figure 6 molecules-27-07621-f006:**
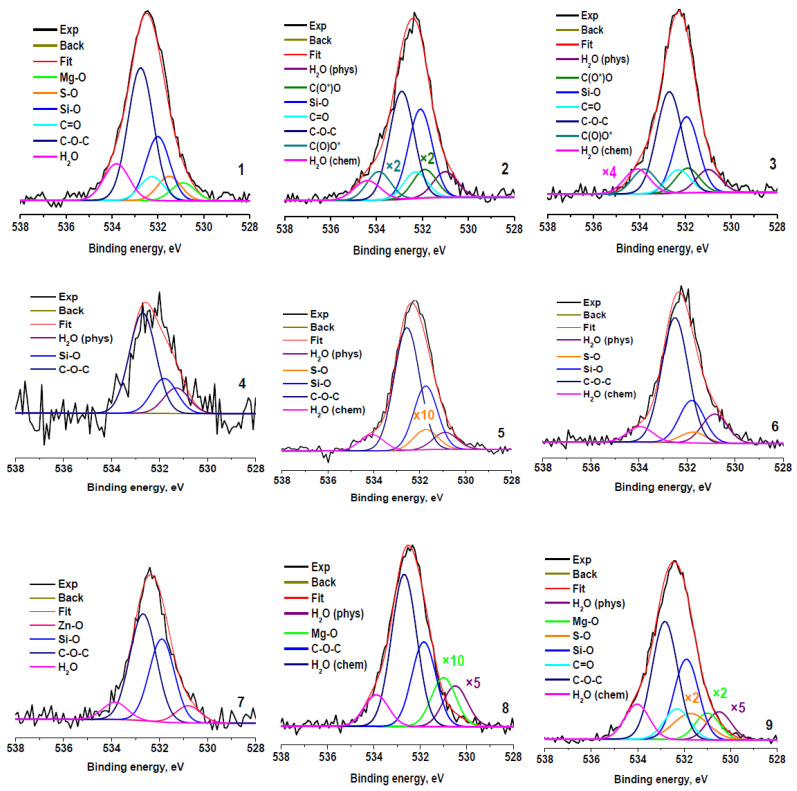
O 1s photoelectron spectra of rubber crumbs, measured in different areas: RC 0-1—**a** (**1**), **b** (**2**), **c** (**3**); RC 1-3—**a** (**4**), **b** (**5**), **c** (**6**); RC 3-5—**a** (**7**), **b** (**8**), **c** (**9**).

**Table 1 molecules-27-07621-t001:** Results of wettability tests for rubber crumbs.

No of Exp.	Sample	Mass of Test Portion of Crumb, g	Mixing Time	Mass of Floated Part, g (% of Reference Mass)	Mass of Settled Part, g (% of Reference Mass)
1	RC 0-1	91	30 min	36 (39.6%)	55 (60.4%)
2	RC 0-1	97	30 min + 12 h	16 (16.5%)	81 (83.5%)
3	RC 1-3	100	5 min	6 (6%)	94 (94%)
4	RC 1-3	100	5 min	7 (7%)	93 (93%)
5	RC 3-5	59	5 min	5 (8.5%)	54 (91.5%)
6	RC 5-10	100	5 min	8 (8%)	92 (92%)

**Table 2 molecules-27-07621-t002:** XPS surface elemental composition of rubber crumb samples (at%).

Sample	C	O	Mg	Si	Zn	S	O (Si)	O (Mg)	S (Zn)	O/C
RC 0-1a	91.5	5.2	0.3	1.1	0.5	1.4	4.2	3.8	0.9	0.06
RC 0-1b	91.3	5.5		1.8	0.5	1.0	5.0	5.0	0.5	0.06
RC 0-1c	91.0	5.7		1.7	0.6	1.0	4.0	4.0	0.4	0.06
RC 1-3a	99.0	0.8		0.2			0.5	0.5		0.01
RC 1-3b	91.8	5.4		1.7	0.5	0.7	3.7	3.8	0.2	0.06
RC 1-3c	95.8	3.1		0.7	0.2	0.3	2.4	2.4	0.1	0.03
RC 3-5a	94.4	3.8		1.5	0.3		2.3	2.3		0.04
RC 3-5b	92.9	4.7	0.1	1.6	0.3	0.5	3.1	3.1	0.2	0.05
RC 3-5c	90.6	6.0	0.3	1.8	0.3	1.0	4.2	3.9	0.7	0.07
RC 3-5 *	90.8	6.0		2.3	0.4	0.5				0.07

* Extracted with chloroform.

**Table 3 molecules-27-07621-t003:** Parameters of components in C 1s photoelectron spectra of rubber crumbs: E_b_—binding energy, W—peak width, and I_rel_—relative intensity. O/C is oxygen to carbon ratio calculated from XPS data.

Sample		C−Si	C−C/C−H	C−S	C−O−C	C=O	C(O)O	O/C
RC 0-1a	E_b_	284.2	285.0		286.0	287.4		
	W	1.03	1.03		1.03	1.03		
	I_rel_	0.07	0.80		0.13	0.01		0.08
RC 0-1a *	E_b_	284.2	285.0	285.6	286.1			
	W	1.03	1.03	1.03	1.03			
	I_rel_	0.07	0.79	0.03	0.11			0.06
RC 0-1b	E_b_	284.1	285.0		286.0	287.1	289.1	
	W	1.03	1.03		1.04	1.03	1.10	
	I_rel_	0.04	0.76		0.17	0.02	0.01	0.12
RC 0-1c	E_b_	284.2	285.0		286.1	287.9	289.0	
	W	1.03	1.03		1.03	1.03	1.10	
	I_rel_	0.08	0.82		0.09	0.01	0.01	0.08
RC 1-3a	E_b_		285.0		286.0			
	W		1.02		1.03			
	I_rel_		0.92		0.08			0.04
RC 1-3b	E_b_	284.1	285.0		286.0			
	W	1.02	1.03		1.03			
	I_rel_	0.06	0.87		0.07			0.04
RC 1-3c	E_b_	283.4	285.0		286.0			
	W	1.03	1.03		1.03			
	I_rel_	0.02	0.91		0.07			0.04
RC 3-5a	E_b_	284.2	285.0		286.0			
	W	1.17	1.17		1.17			
	I_rel_	0.05	0.84		0.11			0.06
RC 3-5b	E_b_	284.2	285.0		286.0			
	W	1.0	1.15		1.11			
	I_rel_	0.06	0.83		0.11			0.06
RC 3-5c	E_b_	284.2	285.0		286.0	287.1		
	W	1.09	1.09		1.09	1.09		
	I_rel_	0.03	0.73		0.21	0.03		0.14
RC 3-5c *	E_b_	284.2	285.0	285.6	286.0	287.1		
	W	1.09	1.09	1.09	1.09	1.09		
	I_rel_	0.07	0.70	0.04	0.16	0.03		0.11

* taking into account the C–S bond.

**Table 4 molecules-27-07621-t004:** Parameters of components in S 2p_3/2_ photoelectron spectra of rubber crumbs: E_b_—binding energy, W—peak width, and I_rel_—relative intensity.

Sample		Zn-S	C−S	C-SO_x_
RC 0-1a	E_b_	162.0	163.7	169.1
	W	1.3	1.35	1.3
	I_rel_	0.50	0.41	0.09
RC 0-1b	E_b_	162.3	163.9	
	W	1.3	1.35	
	I_rel_	0.58	0.42	
RC 0-1c	E_b_	162.3	163.7	
	W	1.3	1.3	
	I_rel_	600	500	
RC 1-3b	E_b_	162.0	163.7	169.1
	W	1.3	1.3	1.3
	I_rel_	0.55	0.32	0.13
RC 1-3c	E_b_	161.9	163.4	169.1
	W	1.24	1.3	1.3
	I_rel_	0.38	0.43	0.19
RC 3-5b	E_b_	162.2	163.8	
	W	1.3	1.3	
	I_rel_	0.47	0.53	
RC 3-5c	E_b_	162.1	163.7	169.1
	W	1.3	1.3	1.3
	I_rel_	0.45	0.41	0.14

**Table 5 molecules-27-07621-t005:** Parameters of components in O 1s photoelectron spectra of rubber crumbs: E_b_—binding energy, W—peak width, and I_rel_—relative intensity.

Sample		H_2_O_ph_	ZnO	MgO	S-O	C(O *)O	Si-O	C=O	C-O-C	C(O)O *	H_2_O_ch_	H_2_O_t_
RC 0-1a	E_b_			530.9	531.5		532.0	532.3	532.8		534.0	
	W			1.05	1.03		1.03	1.05	1.05		1.1	
	I_rel_			0.06	0.08		0.21	0.08	0.45		0.13	0.13
RC 0-1b	E_b_	531.0				531.9	532.1	532.3	532.9	533.9	534.4	
	W	1.05				1.03	1.03	1.03	1.1	1.03	1.15	
	I_rel_	0.06				0.05	0.30	0.09	0.38	0.05	0.07	0.13
RC 0-1c	E_b_	531.0				531.9	532.0	532.3	532.7	533.8	534.1	
	W	1.05				1.03	1.05	1.03	1.05	1.03	1.15	
	I_rel_	264				279	879	264	1230	279	75	
		0.08				0.09	0.27	0.08	0.38	0.09	0.02	0.1
RC 1-3a	E_b_	531.3					531.8		532.7			
	W	1.05					1.05		1.1			
	I_rel_	0.15					0.21		0.63			0.15
RC 1-3b	E_b_	530.9			531.7		531.7		532.6		534.1	
	W	1.05			1.05		1.03		1.1		1.05	
	I_rel_	0.08			0.01		0.28		0.57		0.07	0.08
RC 1-3c	E_b_	530.9			531.7		531.8		532.5		534.0	
	W	1.05			1.05		1.03		1.1		1.05	
	I_rel_	0.13			0.05		0.18		0.57		0.07	0.13
RC 3-5a	E_b_		530.8				531.9		532.7		533.9	
	W		1.03				1.04		1.1		1.04	
	I_rel_		0.02				0.85		0.12		0.02	0.02
RC 3-5b	E_b_	530.5		531.0			531.9		532.7		533.9	
	W	1.05		1.05			1.03		1.08		1.05	
	I_rel_	0.03		0.02			0.29		0.55		0.11	0.14
RC 3-5c	E_b_	530.5		531.0	531.7		531.9	532.3			534.0	
	W	1.05		1.05	1.05		1.05	1.05			1.05	
	I_rel_	0.03		0.07	0.10		0.43	0.17			0.19	0.22

* indicates the corresponding oxygen atom.

## Data Availability

The data presented in this study are available on request from the corresponding author.

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
