# Peer review of "Chemical Composition of Nanoglobular Material on the Surface of Rubber Regenerate Prepared by Explosive Circulation Technology"

_molecules, 2022, doi:10.3390/molecules27217621_

Round 1

Reviewer 1 Report

The paper is dedicated entirely to the chemical composition analysis of automobiles tire crumbs after explosive circular destruction with the purpose to investigate their potential for recycling. Authors have conducted a thorough study and provided detailed interpretation of the results. Even though the study of the composition is limited solely to one technique (XPS), I believe the paper contains some potentially valuable data and merits publication upon minor correction (see below).

1) The language is in good order, save for a few minor instances I managed to spot during reading (e.g. Page 1. “…burred…”  - wrong spelling;  Page 10. Last sentence . “does not settled…”  - incorrect grammar etc.). In this regard, another round of thorough spellcheck is advised.

2) Furthermore, I think authors should consider adding appropriate references to several statements made in the following paragraph on page 2.

“The rubber crumbs produced by explosive milling showed a good adhesion to polymer matrices superior to that of rubber crumbs prepared by mechanical grinding. The former rubber crumbs are used as fillers in various matrices, which allow producing high strength composites. The promising properties of the rubber crumbs produced by explosive milling can be explained by the presence of polar groups on its surface. Therefore, it is important to investigate the wettability of the crumb surface coated with nanoglobules and its chemical composition”.

Author Response

Manuscript ID: molecules-1919988

Title: Chemical composition of nanoglobular material on surface of rubber
regenerate prepared by explosive circulation technology

Authors: Alexander Vasylievich Naumkin *, Vyacheslav Mikhailovich Misin,
Konstantin Igorevich Maslakov

Reviewer 1

The paper is dedicated entirely to the chemical composition analysis of automobiles tire crumbs after explosive circular destruction with the purpose to investigate their potential for recycling. Authors have conducted a thorough study and provided detailed interpretation of the results. Even though the study of the composition is limited solely to one technique (XPS), I believe the paper contains some potentially valuable data and merits publication upon minor correction (see below).

1) The language is in good order, save for a few minor instances I managed to spot during reading (e.g. Page 1. “…burred…”  - wrong spelling;  Page 10. Last sentence . “does not settled…”  - incorrect grammar etc.). In this regard, another round of thorough spellcheck is advised.

Response: The text has been corrected

2) Furthermore, I think authors should consider adding appropriate references to several statements made in the following paragraph on page 2.

“The rubber crumbs produced by explosive milling showed a good adhesion to polymer matrices superior to that of rubber crumbs prepared by mechanical grinding. The former rubber crumbs are used as fillers in various matrices, which allow producing high strength composites. The promising properties of the rubber crumbs produced by explosive milling can be explained by the presence of polar groups on its surface. Therefore, it is important to investigate the wettability of the crumb surface coated with nanoglobules and its chemical composition”.

Response: The text has been corrected. We have added some references.

Good performance characteristics of rubber crumb obtained by explosive grinding were reported in an earlier article by Misin et al. [14]. In addition, information has been added to the Introduction on the expediency of fixing polar hydrophilic groups on the surfaces of various materials in order to improve their properties.

Reviewer 2 Report

The manuscript characterized the chemical compositions of nanoglobular on the surface of rubbers processed through the explosive circular destruction or grinding from worn-out or waste automobile tires studied by XPS. Elemental and chemical compositions are evaluated from XPS spectra on samples condensed and filtered at the bottom of the water to regenerate/recycle the materials effectively. However, XPS data and its analysis are insufficient to explain the high hydrophilicity demonstrated in the water filtering and compatibility with the other materials. The components including minor additives and cords detected in XPS should be consistently compared with the elemental and chemical compositions of the original product before and after the worn-out process under control. It might be inappropriate to use the XPS technique to analyze the chemical compositions on insulated and complex molecules even though the charge neutralization is operated. The C-O/C=O/O-H and their derivatives play a crucial role in the analyses of hydrophilicity and polar groups. However, it is difficult to extract the O1s information as the authors stated. XPS data may be used as a supplementary or reference for the other characterization, but it does not provide the key results related to what the authors concluded. The other minor points are listed below.

- O1s should be analyzed to support the analysis of C-O/C=O bonds observed in C1s.
- The spectra should be displayed with the background because it might tell the inelastic scattering of the photoelectron. The active background method might improve the inconsistency of C/O ratios.
- The atomic % should not provide the C/O "bond" ratios, so the chemical compositions should be derived only from the C1s fitting.
- Adventitious carbon on the surface should be taken into account if the chemical compositions are analyzed ex-situ.
- Explosive milling method should be briefly described and elemental compositions of explosives used should be provided.
- Spectra in Fig. 1 should display clear labels and legends of raw and synthetic peaks. Each figure should be scaled appropriately.
- How do you identify the absence of sp2 from C KVV spectra?
- Optical images should be appropriately analyzed and displayed.
- RC3-5c* sample should be explained in the experimental section.
- Is there any linear combination possible in C KVV spectra as the authors stated the profile in between two spectra?
- S2p should have spin splitting even though it might be small enough to be fitted.

Author Response

Manuscript ID: molecules-1919988

Title: Chemical composition of nanoglobular material on surface of rubber
regenerate prepared by explosive circulation technology

Authors: Alexander Vasylievich Naumkin *, Vyacheslav Mikhailovich Misin,
Konstantin Igorevich Maslakov

Reviewer 2

Comment: The manuscript characterized the chemical compositions of nanoglobular on the surface of rubbers processed through the explosive circular destruction or grinding from worn-out or waste automobile tires studied by XPS. Elemental and chemical compositions are evaluated from XPS spectra on samples condensed and filtered at the bottom of the water to regenerate/recycle the materials effectively. However, XPS data and its analysis are insufficient to explain the high hydrophilicity demonstrated in the water filtering and compatibility with the other materials. The components including minor additives and cords detected in XPS should be consistently compared with the elemental and chemical compositions of the original product before and after the worn-out process under control. It might be inappropriate to use the XPS technique to analyze the chemical compositions on insulated and complex molecules even though the charge neutralization is operated. The C-O/C=O/O-H and their derivatives play a crucial role in the analyses of hydrophilicity and polar groups. However, it is difficult to extract the O1s information as the authors stated. XPS data may be used as a supplementary or reference for the other characterization, but it does not provide the key results related to what the authors concluded. The other minor points are listed below.

Response: This work is a continuation of the work done earlier by one of the co-authors [14. Misin, V.M.; Buryak, A.K.; Zolotarevskii, V.I.; Krivandin, A.V.; Misharina, T.A.; Nikulin, S.S.; Tarasov, A.E. Specifics of the surface of tire crumb regenerate produced by the explosive circulation method. Prot. Met. Phys. Chem. Surf.  2019, 55, 1256–1262.], in which rubber crumb was analyzed by such methods as AFM, X-ray diffraction, gas chromatography, mass spectrometry, and gel-permeation chromatography (GPC).

Comment:  O 1s should be analyzed to support the analysis of C-O/C=O bonds observed in C1s.

Response: Discrimination between C-O and C=O groups in the C 1s spectra does not cause difficulties due to the large difference in chemical shifts 1.13-1.75 eV and 2.81-2.87 eV, respectively [50. Beamson, G.; Briggs, D. High resolution XPS of organic polymers: The Scienta ESCA300 database; Wiley: Chichester [England]; New York, 1992; ISBN 978-0-471-93592-6.]. Usually, fitting the C 1s spectra to discriminate between C-O and C=O groups is more reliable then that of the O 1s spectra. Besides, the O 1s spectra of our samples are rather broad and featureless. Their reliable fitting seems to be impossible because too many oxygen states (components in the spectra) are expected. Along with carbon bonded oxygen a strong contribution from metal oxides/hydroxides and silicon is possible. Therefore, the O1s spectra represent a set of strongly overlapped components. Fitting of such spectra would obviously be overinterpretation.

Comment:  The spectra should be displayed with the background because it might tell the inelastic scattering of the photoelectron. The active background method might improve the inconsistency of C/O ratios. http://www.qro.cinvestav.mx/~aherrera/reportesInternos/activeBackground.pdf

Response: Now, the spectra are displayed with the background and some data in Table 3 were corrected. On the other hand, a very little difference in the background across the C 1s spectra is observed in our samples. In this case even a linear sloping would give a reasonably accurate background subtraction (see, for example, J. Vac. Sci. Technol. A 38, 063203 (2020); https://doi.org/10.1116/6.0000359).

Comment:  The atomic % should not provide the C/O "bond" ratios, so the chemical compositions should be derived only from the C1s fitting.

Response: Table 2 does not present the O/C bond ratios, but the O/C elemental ratios, and they have the right to be presented. By comparing the values obtained using Atomic sensitivity factors (ASFs) and fitting the C 1s spectra, we wanted to show that the former are less reliable because, to the best of our knowledge, the ASFs included in the software of ALL spectrometers in the world do not take into account matrix effects (others in other words, the dependence of the inelastic mean free path on the composition).

Comment:  Adventitious carbon on the surface should be taken into account if the chemical compositions are analyzed ex-situ.

Response: We realize that adventitious carbon can contribute and skew results. However, it is difficult to take into account its contribution, since both adventitious carbon and the studied material itself mainly consist of sp3 carbon. Since the studied samples were prepared under the same conditions, its effect can be considered approximately the same. In this work, it is not the absolute concentrations on the surface that are important, but the difference in composition between the samples. On the other hand, we studied the surface of real, not idealized samples, which will be used in subsequent technological processes.

Comment:  Explosive milling method should be briefly described and elemental compositions of explosives used should be provided.

Response: The text about the essence of the technological process of tire destruction has been added to the experimental part.

Comment:   Spectra in Fig. 1 should display clear labels and legends of raw and synthetic peaks. Each figure should be scaled appropriately.

Response: The Fig. 1 has been changed.

Comment:   How do you identify the absence of sp2 from C KVV spectra?

Response: By using the spectra of polyethylene (sp3) and graphite (sp2). It is impossible to fit the spectrum of graphite into the spectrum of the studied samples.

Comment: Optical images should be appropriately analyzed and displayed.

Response: Reference [14] has been added to the manuscript, in which images of the same fractions are given and analyzed.

Comment: RC3-5c* sample should be explained in the experimental section.

Response: Sample RC3-5c* is described in more detail. The corresponding insert is included in the experimental part. In addition, a change was made to the commentary to the table. 3.

Comment: Is there any linear combination possible in C KVV spectra as the authors stated the profile in between two spectra?

Response: It is evident from Fig. 2 that in our case a sp2-state is not observed in the C KVV spectra. We have mentioned that the C 1s spectra of the rubber crumb samples (Fig. 2) do not show satellites typical for aromatic carbons. In general, linear combination in C KVV spectra was demonstrated by Dementjev et al. (Dementjev A. P., Petukhov, M. N., & Baranov, A. M. (1998). The unique capability of X-ray photon spectroscopy and X-ray excited Auger electron spectroscopy in identifying the sp2/sp3 ratio on the surface of growing carbon films. Diamond and related materials,7(10), 1534-1538)., Kozakov et al. (Kozakov, A. T., Kochur, A. G., Kumar, N., Panda, K., Nikolskii, A. V., & Sidashov, A. V. (2021). Determination of sp2 and sp3 phase fractions on the surface of diamond films from C1s, valence band X-ray photoelectron spectra and CKVV X-ray-excited Auger spectra. Applied Surface Science536, 147807.)

Comment:  S 2p should have spin splitting even though it might be small enough to be fitted.

Response: The S 2p spectra have been corrected

Round 2

Reviewer 2 Report

The reviewer still cannot accept the manuscript for publication.
1. A high hydrophilicity is not explained or analyzed in the XPS results. To analyze the high hydrophilicity, the COOH bond should be identified and evaluated in the XPS data.
2. O1s should be analyzed for COOH and consistently checked with C1s relevant results. S2p is analyzed even though it is smaller than O1s. If O1s has no C-O or C=O, the C1s analysis on C-O, C=O, etc. cannot be sound. If authors follow an XPS fitting procedure (constraint in FWHM), authors can fit each bond in O1s.
3. Elemental compositions are recommended to be compared with the other bulk-sensitive techniques consistently because the XPS results might be modified with the surface contamination. Even though atomic and molecular oxygen or oxygen bonded with the other elements is not considered in the discussion because C and O are principal components, the comparison should be performed with the elemental compositions analyzed by the other techniques such as EDX, ICPMS, etc.

Author Response

  1. A high hydrophilicity is not explained or analyzed in the XPS results. To analyze the high hydrophilicity, the COOH bond should be identified and evaluated in the XPS data.

Response: We have corrected the text after analysis the O 1s spectra which evidence on a presence of water under ultra-high vacuum.

  1. O1s should be analyzed for COOH and consistently checked with C1s relevant results. S2p is analyzed even though it is smaller than O1s. If O1s has no C-O or C=O, the C1s analysis on C-O, C=O, etc. cannot be sound. If authors follow an XPS fitting procedure (constraint in FWHM), authors can fit each bond in O1s.

Response: The O1s spectra were fitted in accordance with the quantification data and fitting the C 1s spectra. A good correspondence is observed.

  1. Elemental compositions are recommended to be compared with the other bulk-sensitive techniques consistently because the XPS results might be modified with the surface contamination. Even though atomic and molecular oxygen or oxygen bonded with the other elements is not considered in the discussion because C and O are principal components, the comparison should be performed with the elemental compositions analyzed by the other techniques such as EDX, ICPMS, etc.

Response: Analysis of the samples by bulk-sensitive methods does not seem appropriate, since the occurrence of any polar groups (including oxygen-containing ones) in the bulk of rubber crumb is unlikely. In addition, even if they occur, they will not affect the wettability and adhesion characteristics of the crumb in any way.